

# Prognostic implication of dynamic platelet count in lung cancer patients with thrombocytosis: a retrospective analysis

Xiaoying Wang, Xiaolin Pu, Xinyu Wu, Yanshuang Wei, Feifei Wei, Fangfang Wu and Hua Jiang

Department of Oncology, The Second People's Hospital of Changzhou, The Third Affiliated Hospital of Nanjing Medical University, Changzhou, Jiangsu, China

## ABSTRACT

**Background.** Thrombocytosis is associated with poor prognosis in lung cancer patients. However, no studies have further assessed the effect of platelet-related parameters on the prognosis among lung cancer patients with thrombocytosis.

**Methods.** Between 2020 and 2021, lung cancer patients with a platelet count $\geq 300$ $*10^9$/L or normal count were retrospectively reviewed. Potential prognostic factors were identified using univariate and multivariate accelerate failure time (AFT) model. Kaplan–Meier method and log-rank test were used to compare survival outcome.

**Results.** Among patients with thrombocytosis ($N = 148$), time point of platelet elevation, platelet distribution width (PDW), platelet-to-lymphocyte ratio (PLR) and neutrophil-to-lymphocyte ratio (NLR) did not significantly impact first-line progression-free survival (PFS). Compared to patients whose platelet count normalized after treatment, patients with sustained platelet elevation exhibited a worst PFS ($\beta = -1.291$, $p < 0.001$), and although patients with fluctuant platelet elevation had worse PFS ($\beta = -0.358$, $p = 0.054$), the difference was not statistically significant. Additionally, mean platelet volume (MPV) ($\beta = 0.319$, $p = 0.008$) and D-dimer ($\beta = -0.046$, $p = 0.025$) were also factors affecting first-line PFS.

**Conclusions.** Among platelet-related parameters, besides MPV and D-dimer, the dynamic pattern of platelet count serves as a prognostic marker in lung cancer patients with thrombocytosis.

## INTRODUCTION

Hematogenous metastasis, the predominant route of cancer dissemination, is responsible for the majority of cancer-related mortality (*Ganesh & Massagué, 2021*). However, this metastatic cascade is characterized by inefficiency (*Massagué & Obenauf, 2016*). Most circulating tumor cells (CTCs) succumb to anoikis, blood flow shear stress, and immune elimination (*Tesfamariam, 2016*). Interestingly, a subset of CTCs can exploit platelets to enhance their survival and complete the metastatic cascade, ultimately forming distant metastases (*Haemmerle et al., 2018*; *López, 2021*). Tumor cells activate and bind

Corresponding author
Hua Jiang, czeyjh@njmu.edu.cn

to platelets, forming a protective "platelet cloak". This cloak provides a physical barrier and facilitates cell–cell adhesion, shielding the cells from the damaging effects of blood flow shear stress and promoting resistance to anoikis (*Haemmerle et al., 2017*; *Schlesinger, 2018*). Furthermore, MHC-I molecules expressed on the platelet surface can aid tumor cells in evading immune surveillance (*Placke et al., 2012*). Platelets also contribute to the metastatic process through the release of various growth factors, which not only induces epithelial-mesenchymal transition (EMT) in CTCs, facilitating their adhesion and extravasation through the vascular endothelium, but also promotes angiogenesis within the specific metastatic microenvironment (*Morris, Schnoor & Papa, 2022*; *Wang et al., 2022*). Therefore, platelets play a critical role in supporting hematogenous metastasis.

The mechanism by which platelets facilitate hematogenous metastasis is strongly corroborated by clinical observations. Tumor cells can induce platelet production through the secretion of interleukin-6 (IL-6), thereby further enhancing their invasive and metastatic capabilities (*Li et al., 2024*). Clinically, elevated platelet counts have been documented in various solid tumors and increased platelet levels have been linked to poor prognosis in cancer patients (*Shu et al., 2024*). Specifically, in patients with non-small cell lung cancer (NSCLC), pre-treatment thrombocytosis is significantly correlated with decreased overall survival (*Sandfeld-Paulsen, Aggerholm-Pedersen & Winther-Larsen, 2023*). Furthermore, in colorectal, lung, ovarian, and gastric cancer patients, elevated platelet counts during diagnosis are linked to higher tumor-specific mortality (*Giannakeas et al., 2022*).

However, previous studies have primarily focused on platelet count at a certain point such as diagnosis, pre-operation, or pre-treatment, while the pro-tumorigenic effects of platelets likely persist throughout the entire process of tumor invasion and metastasis (*Li et al., 2024*). Lung cancer is the malignant tumor with the highest morbidity and mortality in China and patients may exhibit elevated platelets during various stages of treatment. Besides, platelets present with dynamic variability after treatment, manifesting in patterns such as sustained elevation, gradual decline, or an initial drop followed by a subsequent rise. Can the distinct time points of elevation and varying patterns further indicate the prognosis? In addition to the count, platelet-related parameters also include D-dimer, the markers of platelet activation (MPV and PDW) and inflammation (PLR and NLR). Here, we first retrospectively described the distribution characteristics of platelet elevation time points and variation patterns, investigated the impact of these parameters on the prognosis and found that the dynamic pattern of platelet count may serve as a prognostic marker for lung cancer patients with thrombocytosis, which may lay the foundation for clinical studies of anti-platelet therapy.

## MATERIALS & METHODS

### Patient selection

We retrospectively analyzed 1,077 lung cancer patients treated at Changzhou Second People's Hospital from January 2020 to December 2021. Elevated platelets were defined as counts $\geq 300*10^9$/L at any point during diagnosis or treatment, while counts consistently $<300*10^9$/L were considered normal. Patients with acute infections, acute cardio-cerebrovascular diseases, receiving platelet-raising treatment, secondary lung malignancy,

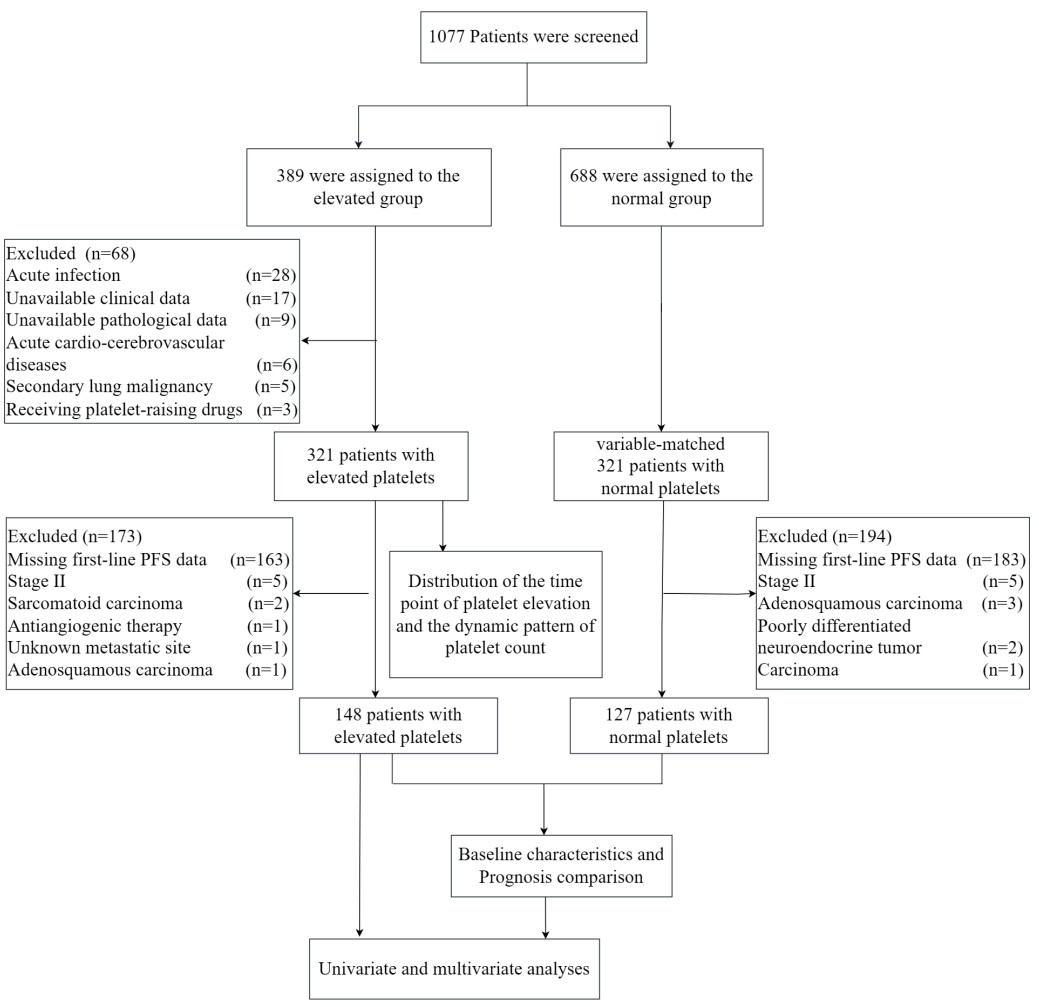

**Figure 1** **The flow chart of the study.**

or missing data were excluded. We examined the time points of platelet elevation in 321 patients with elevated platelets and analyzed dynamic patterns of platelet count in 194 patients with elevated platelets, including 24 who had surgery with postoperative therapy and 170 receiving various treatment lines. Out of 321 patients with elevated platelets, we excluded 163 with missing first-line PFS data and those with small sample sizes due to factors like unknown metastatic sites, stage II disease, and specific cancer types (For details, see Fig. 1). This left 148 patients for prognosis analysis. Similarly, among patients with normal platelet counts, we excluded 183 with missing first-line PFS data and small sample sizes due to similar factors, resulting in 127 patients being included as the negative control group. The study was conducted according to the guidelines of the Declaration of Helsinki, and approved by the Institutional Review Board of The Second People's Hospital of Changzhou (protocol code [2023]KY314-01). Patient consent was waived due to the retrospective nature of the data collected.

## Data collection and definition

In this study, we collected a comprehensive set of patient characteristics from medical records, including demographics (age and gender), tumor characteristics (pathological type, TNM stage, mutations, and programmed cell death ligand-1 (PDL1) expression), disease extent (site of metastasis), and treatment details (number of systemic therapy lines, treatment regimen, and PFS). The treatment encompass chemotherapy, chemo-based combination therapy (including radiotherapy, anti-angiogenic, and immunotherapy), targeted therapy, targeted-based combination therapy (including radiotherapy, anti-angiogenic, and chemotherapy), as well as immunotherapy with or without anti-angiogenic therapy. Additionally, the following parameters were also assessed, including time point of platelet elevation, dynamic patterns of platelet count throughout treatment (if available), MPV, PDW, lymphocyte count, neutrophil count, PLR, NLR, and D-dimer.

Based on the time points of platelet elevation during the treatment course, patients were categorized into the following groups: diagnosis, first-line treatment, progression after first-line treatment, post-operative period, adjuvant treatment, post-operative recurrence, second-line treatment, progression after second-line treatment, third-line or later treatment, and progression after third-line or later treatment.

Based on the dynamic patterns of platelet count following treatment, three patterns were identified: peak-type (where the elevated platelet count normalized after treatment), fluctuation-type (where the elevated platelet count normalized and then increased again later), and sustained-type (where the platelet count remained elevated throughout treatment).

Tumors were assessed at baseline, and efficacy was evaluated according to the criteria of RECIST 1.1. Progressive disease (PD) was defined as an increase of more than 20% in the sum of the longest diameters of all target lesions or the appearance of one or more new lesions. The primary endpoint was first-line PFS, which is defined as the duration from the initial of first-line treatment to the first occurrence of disease progression or death from any cause. The independent variables of the study include the time point of platelet elevation, dynamic patterns of platelet count, and platelet-related numerical parameters such as MPV, PDW, NLR, PLR and D-dimer. The covariates considered are clinical characteristics of lung cancer patients, including stage, histology, treatment and so on.

## Statistical analysis

This study utilized several statistical techniques for data analysis. Baseline characteristics were assessed using the Wilcoxon rank-sum test for non-normally distributed continuous and ordinal variables, and the Pearson chi-square test for categorical variables. In the univariate analysis, Kaplan–Meier survival curves, along with log-rank tests, were initially employed to compare PFS between patients with elevated and normal platelets, serving as a primary descriptive and comparative measure of survival outcomes. Then both univariate and multivariate regression models were conducted to explore factors influencing patient survival outcomes. An initial exploration of potential variables affecting PFS was conducted using a univariate AFT model. Variables demonstrating statistical significance in the univariate AFT model were subsequently included in next-step multivariate analysis,

wherein a multivariate AFT model was developed. In this multivariate AFT model analysis, a stepwise method was employed to establish the model and identify factors potentially associated with PFS.

In the subgroup analysis, we applied a consistent methodology to patients with elevated platelets to investigate factors associated with PFS. Initially, a univariate AFT model was utilized to identify factors potentially related to PFS. Factors demonstrating statistical significance in the univariate AFT model were subsequently included in a multivariate AFT model analysis. Those factors that remained statistically significant in the multivariate AFT model were ultimately identified as influential determinants of PFS in patients with elevated platelets. All statistical analyses were performed using the R statistical programming language (version 4.3.2), and a $p$-value of less than 0.05 was statistically significant.

# RESULTS

## Baseline characteristics and prognosis of lung cancer patients

Baseline characteristics of patients with elevated and normal platelets are presented in Table 1. There were no significant differences between the two groups in sex, age at diagnosis, lymphocyte count, NLR, histology, clinical stage, and treatment. Data were based on existing records, and not all patients were tested for mutations and PD-L1 expression, so these variables weren't statistically analyzed. The elevated platelet group had a higher proportion of ALK mutations (6% *vs.* 0.7%, respectively), but due to the small sample sizes this requires further confirmation. Metastatic sites were similar in both groups. Compared to patients with normal platelets, those with elevated platelets had significantly lower MPV and PDW, as well as higher neutrophils, PLR, and D-dimer. The follow-up time in the elevated group was significantly shorter, indicating that earlier disease progression was observed.

We initially analyzed prognostic factors in lung cancer patients. Kaplan–Meier survival curves showed that histology and treatment were significantly associated with first-line PFS (Fig. S1). The PFS was significantly lower in males than females, potentially due to a higher mutation prevalence in female patients (65% *vs.* 35%, respectively), and targeted therapy, both mono- and combination regimens, were associated with superior PFS outcomes. Univariate analysis revealed that in addition to sex, histology, and treatment, elevated platelets and D-dimer were closely related to PFS (Table S1). Further multivariate AFT model analysis confirmed that elevated platelets, along with D-dimer and treatment, were correlated with PFS in lung cancer patients (Table S2).

## The distribution of the time point of platelet elevation and the dynamic pattern of platelet count in patients with thrombocytosis

As shown in the flow chart (Fig. 1), 68 patients were first excluded and a total of 1,009 patients with lung cancer were documented. Of these, 321 patients were included in the elevated platelet group, representing 31.8% of the total included patients. Most patients experienced an increase in platelet count at the time of diagnosis (50.47%), followed by the first-line treatment period (20.56%) (Fig. 2). Among the 194 patients who completed treatment and post-treatment follow-up, the peak-type pattern was the most common

**Table 1  Baseline clinical characteristics of lung cancer patients with elevated and normal platelets.**
Others metastatic sites include adrenal gland, pericardium, pancreas, spleen and ovary.

| Characteristic | Elevated platelet group N = 148 | Negative control group N = 127 | P-value[a] |
|---|---|---|---|
| Sex, n (%) | | | 0.289 |
|     Female | 52 (35) | 37 (29) | |
|     Male | 96 (65) | 90 (71) | |
| Age, Mean ± SD | 63.97 ± 10.06 | 65.54 ± 8.63 | 0.367 |
| MPV, Median (IQR) | 9.65 (9.20–10.20) | 10.60 (9.80–11.60) | <0.001 |
| PDW, Median (IQR) | 10.65 (9.80–11.65) | 12.50 (10.90–14.60) | <0.001 |
| Lymph, Median (IQR) | 1.48 (1.14–1.92) | 1.43 (1.14–1.73) | 0.485 |
| NEU, Median (IQR) | 5.32 (3.86–7.14) | 4.22 (3.32–5.30) | <0.001 |
| D-dimer, Median (IQR) | 0.84 (0.46–1.56) | 0.59 (0.29–1.45) | 0.005 |
| NLR, Median (IQR) | 3.41 (2.16–5.91) | 3.04 (2.07–4.09) | 0.056 |
| PLR, Median (IQR) | 240.34 (181.76–314.25) | 137.09 (103.45–178.98) | <0.001 |
| Histology, n (%) | | | 0.362 |
|     AC | 91 (61) | 72 (57) | |
|     NSCLC | 8 (5.4) | 3 (2.4) | |
|     SCC | 22 (15) | 21 (17) | |
|     SCLC | 27 (18) | 31 (24) | |
| Clinical stage, n (%) | | | 0.227 |
|     III | 38 (26) | 25 (20) | |
|     IV | 109 (74) | 102 (80) | |
|     Recurrence | 1 | 0 | |
| Treatment, n (%) | | | 0.254 |
|     CT | 39 (26) | 29 (23) | |
|     CCT | 54 (36) | 47 (37) | |
|     IT ± AT | 5 (3.4) | 3 (2.4) | |
|     TCT | 25 (17) | 14 (11) | |
|     TT | 25 (17) | 34 (27) | |
| Mutation target, n (%) | | | – |
|     Del19 | 25 (17) | 26 (18) | |
|     L858R | 13 (9) | 15 (12) | |
|     ALK | 9 (6) | 1 (0.7) | |
|     ROS1 | 2 (1) | 0 (0) | |
|     Others | 6 (4) | 6 (5) | |
| Metastatic sites, n (%) | | | |
|     Bone | 42 (28) | 44 (35) | |
|     Pleura | 33 (22) | 27 (21) | |
|     Lung | 29 (20) | 20 (16) | |

**Table 1** (*continued*)

| Characteristic | Elevated platelet group N = 148 | Negative control group N = 127 | P-value[a] |
|---|---|---|---|
| Brain | 16 (11) | 13 (10) | |
| Liver | 10 (7) | 11 (9) | |
| Others | 8 (5) | 12 (9) | |
| PDL1>1%, n (%) | 15 (10) | 10 (8) | |
| Follow up (days), Median (IQR) | 296.00 (172.50–495.00) | 330.00 (226.00–589.00) | 0.019 |

**Notes.**

Lymph, Lymphocytes; NEU, Neutrophils; AC, Adenocarcinoma; SCC, squamous cell carcinoma; SCLC, small cell lung cancer; CT, chemotherapy; CCT, chemo-based combination therapy; IT, immunotherapy; AT, antiangiogenic therapy; TT, targeted therapy; TCT, targeted combination therapy, the same below.

[a]Pearson's Chi-squared test; Wilcoxon rank sum test; Fisher's exact test.

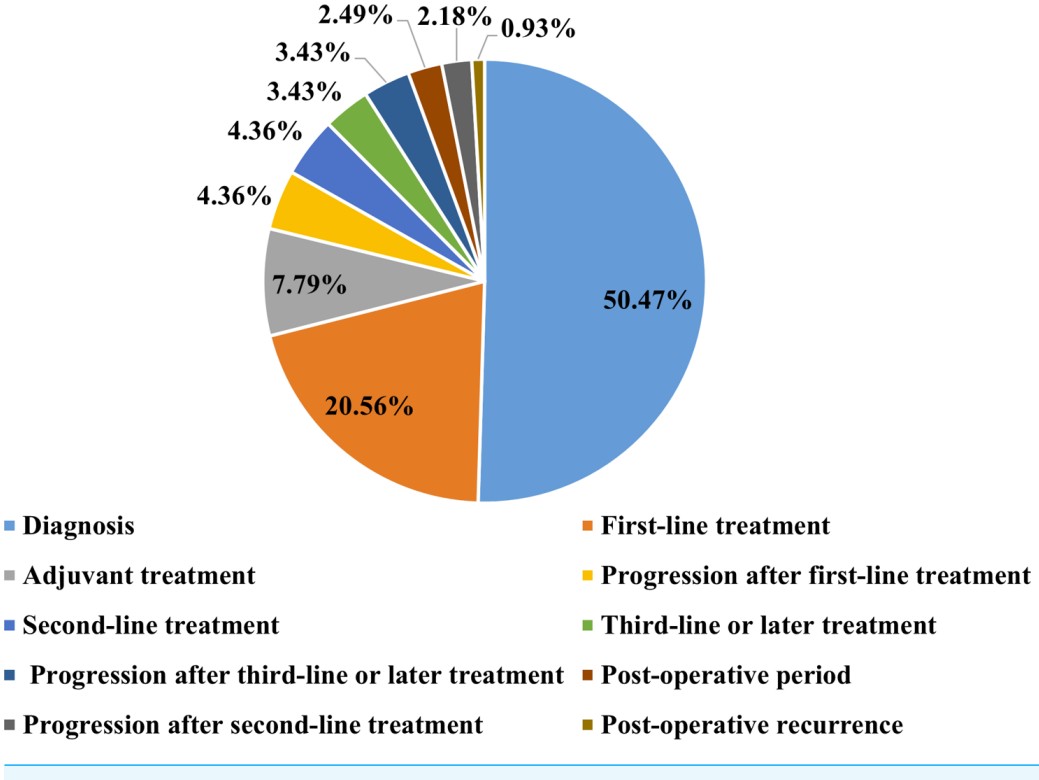

**Figure 2** The distribution of the time points of platelet elevation in lung cancer patients with thrombocytosis (N = 321).

(46.91%), followed by the fluctuation-type (42.78%), and the least prevalent was the sustained-type (10.31%) (Fig. 3).

## Prognostic factors for lung cancer patients with thrombocytosis

Based on the result that elevated platelets were correlated with first-line PFS in lung cancer patients, we further investigated the prognostic factors for lung cancer patients with thrombocytosis. As shown in Fig. 4, the time point of platelet elevation, whether at

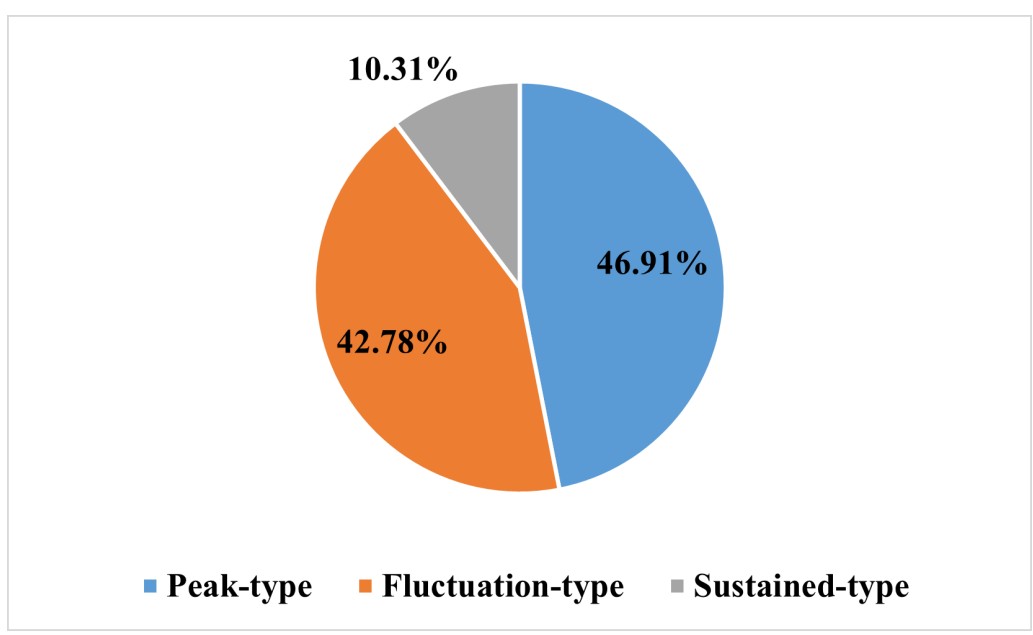

**Figure 3** **The distribution of the dynamic patterns of platelet count in lung cancer patients with thrombocytosis ($N = 194$).**

diagnosis or during first-line treatment, did not associated with first-line PFS ($p = 0.81$). However, dynamic patterns of platelet count showed a significant correlation with first-line PFS ($p < 0.0001$). Univariate analysis revealed the same general results (Table 2). While there were significant differences in neutrophil and PLR between patients with elevated and normal platelets, these variables did not show a notable association with PFS in patients with elevated platelets. Patients with elevated platelets exhibited lower MPV and PDW and higher D-dimer levels, all of which were significantly associated with first-line PFS.

To further investigate the influence of platelet-related factors, a multivariate analysis was conducted (Table 3). The dynamic pattern of platelet count remained strongly associated with first-line PFS. Compared to patients with peak-type, patients with sustained-type had the worst PFS ($\beta = -1.291$, $p < 0.001$). Although patients with fluctuation-type exhibited worse PFS ($\beta = -0.358$, $p = 0.054$), the difference was not statistically significant. Among numerical variables, MPV was positively correlated with PFS, while D-dimer was negatively correlated.

## DISCUSSION

This study investigated the impact of platelet-related parameters on first-line PFS in lung cancer patients. Patients with elevated platelets exhibited a significantly shorter PFS compared to those with normal platelets. Furthermore, the dynamic pattern of platelet count after treatment emerged as an independent prognostic factor for patients with thrombocytosis.

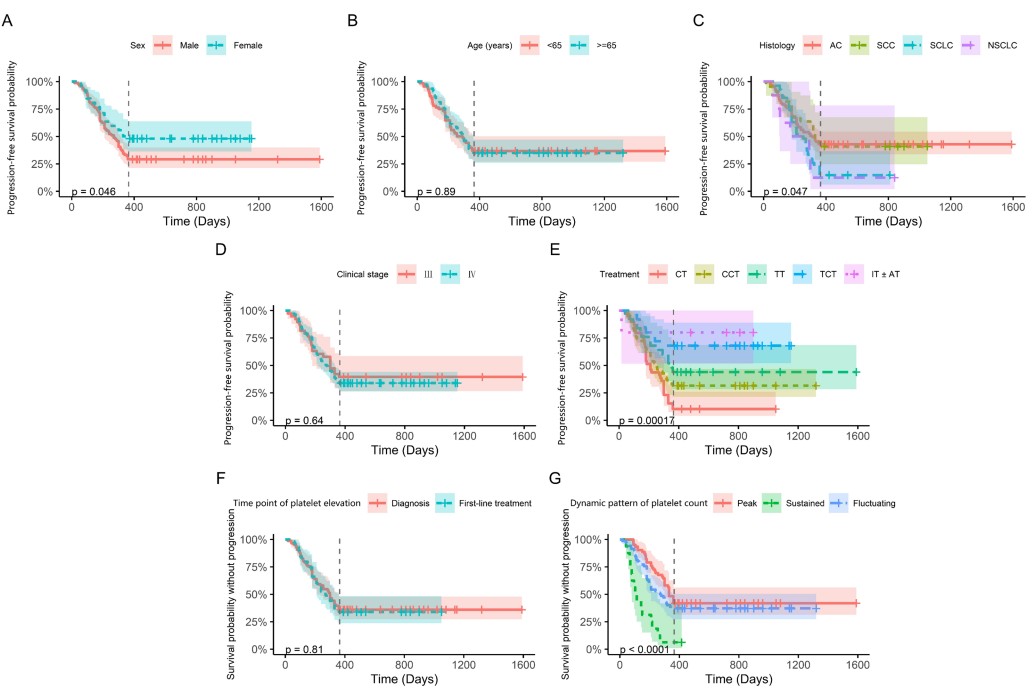

**Figure 4  Kaplan–Meier progression-free survival curves for patients with elevated platelets.** The vertical line represents time=365 days, which is the 1-year follow-up point. Panels (A) to (G) display the analyzed variables, including sex, age, histology, clinical stage, treatment, time point of platelet elevation, and dynamic pattern of platelet count, respectively.

Elevated pretreatment platelet counts are associated with a poor prognosis in lung cancer patients (*Li et al., 2024*; *Sandfeld-Paulsen, Aggerholm-Pedersen & Winther-Larsen, 2023*). However, thrombocytosis can occur at any stage of lung cancer. This study aims to investigate whether elevated platelet counts occurring during treatment have the same prognostic impact as those observed before treatment. Besides, lung cancer patients with elevated platelets exhibit various patterns after treatment. Some patients' platelet counts drop to normal levels, while others experience an initial drop followed by a rise, and yet others experience a continuous rise. The different point of platelet elevation and the varying patterns of platelet count changes may provide additional prognostic information. Existing studies are limited to a single time point of platelet elevation, and no research has examined the dynamic changes in platelet counts after treatment.

To address the limitations, a dynamic follow-up study was conducted in lung cancer patients. For patients with unresectable stage III and stage IV disease, PFS did not differ significantly between those with elevated platelets at diagnosis and those with elevated platelets during first-line therapy. This suggests that the time point of platelet elevation may not be the main factor affecting prognosis. Patients with elevated platelets exhibited a reduced PFS compared to those with normal platelets. Furthermore, the dynamic patterns in platelet count after treatment provided additional prognostic information. Compared to patients with the peak-type, those with the sustained-type exhibited the shortest PFS.

**Table 2  Regression coefficients of the univariable AFT model for first-line PFS in lung cancer patients with thrombocytosis.**

| Variable | β(95% CI) | SE | Z | P |
|---|---|---|---|---|
| Sex (Female) | 0.44 (−0.014, 0.893) | 0.232 | 1.899 | 0.058 |
| Age | 0.004 (−0.017, 0.025) | 0.011 | 0.379 | 0.704 |
| MPV | 0.394 (0.111, 0.677) | 0.145 | 2.725 | 0.006 |
| PDW | 0.177 (0.025, 0.328) | 0.077 | 2.289 | 0.022 |
| Lymph | −0.042 (−0.359, 0.274) | 0.161 | −0.263 | 0.793 |
| NEU | −0.031 (−0.095, 0.033) | 0.033 | −0.956 | 0.339 |
| D-dimer | −0.061 (−0.109, −0.012) | 0.025 | −2.441 | 0.015 |
| NLR | −0.024 (−0.079, 0.032) | 0.028 | −0.842 | 0.4 |
| PLR | −0.001 (−0.002, 0.001) | 0.001 | −0.744 | 0.457 |
| Histology | | | | |
|    AC | Reference | – | – | – |
|    SCC | 0.091 (−0.528, 0.71) | 0.316 | 0.288 | 0.773 |
|    SCLC | −0.438 (−0.943, 0.067) | 0.258 | −1.700 | 0.089 |
|    NSCLC | −0.662 (−1.516, 0.192) | 0.436 | −1.519 | 0.129 |
| Clinical stage (IV) | −0.159 (−0.654, 0.336) | 0.253 | −0.629 | 0.53 |
| Treatment | | | | |
|    CT | Reference | – | – | – |
|    CCT | 0.296 (−0.157, 0.749) | 0.231 | 1.282 | 0.2 |
|    TT | 0.677 (0.09, 1.263) | 0.299 | 2.260 | 0.024 |
|    TCT | 1.494 (0.819, 2.168) | 0.344 | 4.343 | <0.001 |
|    IT ± AT | 1.944 (0.394, 3.494) | 0.791 | 2.458 | 0.014 |
| Time point of platelet elevation | −0.067 (−0.492, 0.359) | 0.217 | −0.306 | 0.759 |
| Dynamic pattern of platelet count | | | | |
|    Peak | Reference | – | – | – |
|    Sustained | −1.319 (−1.921, −0.718) | 0.307 | −4.300 | <0.001 |
|    Fluctuation | −0.32 (−0.739, 0.1) | 0.214 | −1.495 | 0.135 |

Although patients with fluctuation-type also demonstrated poorer PFS, the *p*-value was 0.054, indicating that a larger sample size is needed to confirm the conclusion.

Dynamic pattern of platelet count can reflect tumor cell response to treatment. Tumor heterogeneity dictates that only a subset of tumor cells can promote platelet production through a positive feedback loop involving IL-6 secretion. The persistence of these cells suggests an insensitive response to antitumor drugs, leading to rapid disease progression and significantly reduced PFS. Conversely, normalization of platelet count after treatment signifies treatment sensitivity, indicating disease control and relatively prolonged PFS. From this perspective, an elevated platelet count can be seen as a marker for the presence and activity of tumors. Thrombocytosis, affecting nearly a third of all lung cancer patients, presents a significant challenge due to its unsatisfactory prognosis, and anti-platelet combined with anti-tumor therapies may offer a promising strategy, especially for sustained-type patients who benefit least from first-line anti-tumor therapy.

**Table 3** Regression coefficients of the multivariable AFT model for first-line PFS in lung cancer patients with thrombocytosis.

| Variable | β(95%CI) | SE | Z | P |
| --- | --- | --- | --- | --- |
| Treatment | | | | |
| CT | Reference | – | – | – |
| CCT | 0.285 (−0.112, 0.681) | 0.202 | 1.407 | 0.16 |
| TT | 0.874 (0.351, 1.397) | 0.267 | 3.277 | 0.001 |
| TCT | 1.268 (0.659, 1.876) | 0.310 | 4.085 | <0.001 |
| IT ± AT | 1.902 (0.534, 3.27) | 0.698 | 2.726 | 0.006 |
| Dynamic pattern of platelet count | | | | |
| Peak | Reference | – | – | – |
| Sustained | −1.291 (−1.833, −0.75) | 0.276 | −4 .676 | <0.001 |
| Fluctuation | −0.358 (−0.721, 0.006) | 0.185 | −1.930 | 0.054 |
| MPV | 0.319 (0.083, 0.555) | 0.120 | 2.652 | 0.008 |
| D-dimer | −0.046 (−0.086, −0.006) | 0.021 | −2.238 | 0.025 |

Interestingly, patients with the sustained-type exhibited the poorest prognosis, even though the cumulative duration of platelet elevation in a part of patients with the fluctuation-type and peak-type was longer. This observation suggests that the sustained nature of platelet elevation, rather than the total time of elevated platelets, has a greater impact on prognosis. This finding potentially reflects the dominant role of tumor cells in the processes of cancer invasion and metastasis.

Increased MPV and PDW are often used as markers of platelet activation (*Feng et al., 2019*). This study observed that patients with elevated platelets exhibited significantly lower MPV and PDW compared to patients with normal platelets. Meantime, MPV was positively correlated with PFS, while platelet count was negatively correlated. These results suggest that platelet activation may not necessarily accompany an increase in platelet count. MPV exhibits an inverse correlation with platelet count, which are likely to be dynamically balanced through negative feedback regulation within the bone marrow. The PLR and NLR are inflammatory markers commonly used to assess the inflammatory state in cancer patients, which is believed to contribute to tumorigenesis and disease progression (*Ma et al., 2022*). Our study found that neither PLR nor NLR significantly impacted PFS in patients with elevated platelets. This inconsistency may be due to variations in the specific numerical cutoffs, sample size limitations, tumor heterogeneity and so on.

Platelets contribute to a hypercoagulable state in cancer patients, indicated by high D-dimer levels, which predict venous thromboembolism and are linked to tumor metastasis and mortality (*Ay et al., 2012*). Our study confirmed that patients with elevated platelets have higher D-dimer levels, which correlate with shorter PFS. In this context, anti-platelet therapy may reduce the risk of tumor metastasis and mitigate hypercoagulability, potentially offering a promising adjunct strategy for cancer treatment.

Despite the novel findings of this study, several limitations warrant consideration. First, the study's retrospective design and reliance on data from a single institution limit the generalizability, and the relatively small sample size may restrict the power of the

statistical analysis. Second, the absence of certain clinical data and the limited follow-up time precluded the collection and analysis of overall survival (OS) data. To further validate these findings, future studies should involve multiple centers, a larger sample size, and a prospective design.

## CONCLUSIONS

In summary, we evaluated the prognostic significance of platelet and platelet-related parameters in lung cancer patients and found that the dynamic pattern in platelet count may serve as a prognostic marker, with the poorest prognosis observed in patients with sustained elevated platelets. Our findings may lay the groundwork for clinical studies of anti-platelet treatment in lung cancer patients.

### Funding

This work was funded by the Changzhou Sci&Tech Program (Grant No. CJ20230065). The funders had no role in study design, data collection and analysis, decision to publish, or preparation of the manuscript.

### Grant Disclosures

The following grant information was disclosed by the authors:
Changzhou Sci&Tech Program: CJ20230065.

### Competing Interests

The authors declare there are no competing interests.

### Author Contributions

- Xiaoying Wang conceived and designed the experiments, performed the experiments, authored or reviewed drafts of the article, and approved the final draft.
- Xiaolin Pu conceived and designed the experiments, performed the experiments, authored or reviewed drafts of the article, and approved the final draft.
- Xinyu Wu performed the experiments, prepared figures and/or tables, and approved the final draft.
- Yanshuang Wei performed the experiments, prepared figures and/or tables, and approved the final draft.
- Feifei Wei analyzed the data, prepared figures and/or tables, and approved the final draft.
- Fangfang Wu analyzed the data, prepared figures and/or tables, and approved the final draft.
- Hua Jiang conceived and designed the experiments, performed the experiments, authored or reviewed drafts of the article, and approved the final draft.

### Human Ethics

The following information was supplied relating to ethical approvals (i.e., approving body and any reference numbers):

The Institutional Review Board of The Second People's Hospital of Changzhou (protocol code [2023]KY314-01).

## Data Availability

The raw data are available in the Supplementary File.

## Supplemental Information

Supplemental information for this article can be found online at http://dx.doi.org/10.7717/peerj.19551#supplemental-information.

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
