# Peer review of "Prognostic implication of dynamic platelet count in lung cancer patients with thrombocytosis: a retrospective analysis"

_PeerJ, doi:10.7717/peerj.19551_

## Round 0.1 · original submission · Major Revisions

Dear authors,
thank you for your submission. Your manuscript requires significant revisions. Your methodology and statistical analysis in particular requires some attention. Please, refer to the reviewers comments for further details.

Reviewer 1 ·

Basic reporting

Wang et al. propose a study that aims to investigate the role of platelet count during lung cancer treatment, building upon previous studies that had only analyzed this aspect prior to the start of therapy. Based on the literature, platelets play an important role during the metastasis process, linking pre-treatment platelet elevation to poor prognosis.

The study aims not only to evaluate the rise in platelet count, but also other parameters related to platelet activity, as well as the analysis of dynamic changes in platelet values and the timing of their elevation.

The English is clear, and the paper structure is well-organized, with a clear description of the state of the art.

The proposed figures are quite basic and old style.

The description of the cohort analyzed, as well as the parameters studied, is clear.

However, the section on the role of aspirin is somewhat unclear, especially in relation to the fact that there are conflicting studies in the literature regarding its benefits in lung cancer. Moreover, none of the patients in the study cohort were treated with aspirin or this is not clear from Table 3. A study conducted by the same authors is cited; the study observed that tumor cells binding to platelets correlated with a more pronounced inhibitory effect of aspirin. However, the study remains unpublished, and the results are not verifiable. It is unclear why aspirin was included as a theme in the conclusions

Experimental design

the study meet the experimental design

Validity of the findings

It would be nice to add a Kaplan-Meier curve to show the different patterns of platelets counts and how they are related to PFS

I would have investigated the mutational aspect further, especially in relation to the three different patterns of dynamic change of platelet count and how this might influence prognosis. Is there any association between ALK, L858R, Del19 and PFS?

The 194 patients described as having completed treatment—does this refer to first-line treatment? If so, it should be specified

Low levels of MPV are typically associated with chemotherapy, were those patients with low levels of MPV treated with chemotherapy?

In the multivariate analysis, a correction for confounding factors is reported, but which factors were considered? What type of model was used? It’s not clear.

Can you speculate more on why in your study, patients with elevated platelets counts show low MPV and PDW levels?

Additional comments

FINAL CONSIDERATION:
A limitation of the study is its small sample size, as noted by the authors themselves. However, it could be considered a pilot study for evaluating the role of platelets in tumor progression over time.

Reviewer 2 ·

Basic reporting

Wang et al conducted a retrospective analysis of the relationship between thrombocytosis and lung cancer progression. Though the sample size is limited, the authors conducted standard biostatistical analyses, showing that patients with thrombocytosis experienced shorter PFS and dynamic platelet count can serve as a prognostic factor. Some issues can be clarified before publication.

Experimental design

Minor:
1. The authors showed the presence and pattern of platelet elevation is associated with shorter PFS. Reporting the results in the form of Kaplan-Meier curves and CoxPH model is more convincing.
2. It was not written clearly in the method section how the multivariate analyses were done.

Validity of the findings

Major:
1. Based on their analyses, the authors suggest that patients with elevated platelet levels may benefit from aspirin, which is known to have pleiotropic effects. Can authors clarify if patients in their cohort received aspirin treatment? Aspirin treatment status can also be included in the multivariate model.

Reviewer 3 ·

Basic reporting

o The English language is generally clear to understand. But there are some examples where there are typos or unclear presentations:
 Some variable name is confusing: time to platelet count is a time to event variable, defined as the time from start to first occurrence of platelet elevation. However, the authors considered it as a binary variable (at diagnosis, during first-line treatment), which is not correct. it should be renamed to “stage when platelets count is elevated” or something similar.
 Line 82: should be patient selection instead of patient section
 In line 24-25, line 75, line 100, line 170, need to clarify what “/” means in “MPV/PDW” and “PLR/NLR”, “division” or “and”?
o Figure and table titles are unclear or missing
 The title of figure 2 is “the distribution of dynamic change in platelet count”. Please clarify which patient group that the figure is based on.
 Please clarify if figure 1 is for all lung cancer patients.
 Table titles are missing, for example supplementary tables

Experimental design

o Issues with study design: From Line 83, the author mentioned that they conducted a case-control study, with case being the group of patients with elevated platelet count, and control being the group of patients with normal platelet count. However, this is not aligned with the study objective, which is to evaluate the effect of dynamic platelet count/platelet-related parameters on prognosis of lung cancer patients with thrombocytosis.
In fact, the outcome variable (or response variable, dependent variable) is the progression-free survival of patients with abnormal platelet count, rather than whether the platelet count of patients is normal or abnormal. And the exposure factor is the dynamic change in platelet count/platelet related parameters. So it does not make sense to compare patients with elevated platelet count versus patients with normal platelet count. Instead, the analysis should be done to see how dynamic change in platelet count/ platelet parameter affects PFS time in patients with abnormally high number of platelets.

And the definition of patient population should be clarified: since the platelet count may change over time, it should be clarified how to define the patients with elevated platelet count. Are they the patients with platelet count>300x10^9/L at baseline, regardless of how the platelets count change during treatment?

o Data collection: as part of the study design, it should be clarified in “Data collection and definition” section (line 94) the total number of patients selected into the study and how they were selected (eligibility criteria) and the selection rationale. From results section, it looks that 321 patients with elevated platelet were selected from a total of 1009 patients with lung cancer. And then 103 patients with certain characteristics, such as stage III-IV (from line 153) were selected.
o Variable of interest: need to clarify in the “methods” section about which variable is the primary outcome of interest, and which variables are the independent variables that may affect the prognosis of lung cancer patients with thrombocytosis. From the manuscript, the primary outcome of interest is PFS, and other independent variable that may affect PFS including dynamic platelets count and platelet-related parameters
 And the authors should clarify how PFS is defined. PFS is usually defined as time from treatment start to disease progression. Since patients may have multiple disease progressions during the treatment and may have received multiple lines of treatment, patients could have multiple PFS, depending on the start and end time of PFS evaluation.
 It is important to ensure the evaluation period for PFS is consistent with the evaluation period for platelets count as well as for other platelet-related parameters of interest.
o Stats methods:
 Given that the patient population of interest is lung cancer patients with thrombocytosis, comparing different variables between patients with elevated platelets count vs patients with normal platelets count may not be relevant, if what the authors meant by “patients with normal platelets count” was the patients who never had abnormal platelets count from start to end. The authors may present a summary of characteristics of all lung cancer patients, but the focus in “Patient characteristics” (line 127) should be patients with abnormal platelets count.
 One major issue with the analysis is that PFS was considered as a univariate continuous variable instead of a bivariate time-to-event variable with both continuous PFS time and binary censoring status. The observed value of PFS may not necessarily reflect the actual PFS time of patients, due to censoring in which PFS event is not observed. So the author should consider methods/models that can handle time to event variable, such as, Cox model, log-rank test, instead of using simple linear regression. This comment may have affected stats results reported in abstract and the result section.
 It is not clear how the univariate/multivariate analysis of prognostic factors are conducted in patients with elevated platelets count. Please clarify the models used in statistical analysis section.
o General comments: The above comments apply to the main text as well as the abstract: The abstract section should be updated.

Validity of the findings

o Given the comments in experimental design section, the results need to be updated accordingly based on new models.
o Additionally, some minor comments:
 Line 129: is it 7.48% based on table 1?

---

## Round 0.2 · Major Revisions

Dear authors, thank you for your revisions. However, some very relevant questions remain. Please, address them to assure the scientific soundness and relevance of your work. many thanks.

Reviewer 1 ·

Basic reporting

no comment

Experimental design

no comment

Validity of the findings

no comment

Reviewer 2 ·

Basic reporting

The authors answered my questions and I recommend publication.

Experimental design

The authors clarified the models they used,

Validity of the findings

The authors clarified the patient population they studied.

Additional comments

I don't have further questions.

Reviewer 3 ·

Basic reporting

Please see additional comments

Experimental design

Please see additional comments

Validity of the findings

Please see additional comments

Additional comments

In general, the language in the article is well written and the authors responded to comments from the first round of review. However, there are still major concerns regarding statistical analyses. Several statistical analyses were conducted but many of them were not relevant to the stated primary goal of the study. And the method of the primary analysis was written in the way that does not seem to be statistically meaningful. This makes the logic of the paper does not "flow", confusing and hard to follow. In statistical analysis section, the methods described lack sufficient details and information to replicate. For example,
• initially the univariate analysis was used to identify potentially significant variables that impact PFS, then a multivariate model was built. It was then mentioned that a stepwise regression was used to identify prognostic factors associated with PFS. But the results section does not seem to include the results of the stepwise method, or it was not clearly written.
• prognostic factors in lung patients were identified initially, which showed elevated platelet is associated with PFS. Then the authors explored prognostic factors in lung patients with thrombocytosis. The latter was the major scientific question to be address, however, was not described/defined in the statistical methods section.

---

## Round 0.3 · accepted · Accept

Dear authors,

All issues have now seem to have been satisfactorily addressed. Your manuscript is now accepted for publication. Congratulations!

Reviewer 3 ·

Basic reporting

No additional comments

Experimental design

No additional comments

Validity of the findings

No additional comments